

# Intercomparison of the weather and climate physics suites of a unified forecast/climate model system (GRIST-A22.7.28) based on single column modeling

Xiaohan Li[1, 2], Yi Zhang*[1, 2], Xindong Peng[2, 3], Baiquan Zhou[2], Jian Li[2], and Yiming Wang[1]

[1]2035 Future Laboratory, PIESAT Information Technology Co Ltd., China

[2]State Key Laboratory of Severe Weather, Chinese Academy of Meteorological Sciences, Beijing 100081, China

[3]CMA Earth System Modeling and Prediction Center, Beijing, China

*Correspondence to:* Yi Zhang, Email: zhangyi_fz@piesat.cn

**Abstract.** As a unified weather-forecast/climate model system, Global-to-Regional Integrated forecast SysTem (GRIST-A22.7.28) currently employs two separate physics suites for weather forecast and typical long-term climate simulation, respectively. Previous AMIP-style experiments have suggested that the weather (PhysW) and climate (PhysC) physics suites, when coupled to a common dynamical core, lead to different behaviors in terms of modeling clouds and precipitation. To explore the source of their discrepancies, this study compares the two suites using a single column model (SCM). The SCM simulations demonstrate significant differences in the simulated precipitation and low clouds. Convective parameterization is found to be a key factor responsible for these differences. Compared with PhysC, parameterized convection of PhysW plays a more important role in moisture transport and rainfall formation. The convective parameterization of PhysW also better captures the onset and retreat of rainfall events, but stronger upward moisture transport largely decreases the tropical low clouds in PhysW. These features are in tune with the previous 3D AMIP simulations. Over the typical stratus-to-stratocumulus transition regime such as the Californian coast, shallow convection in PhysW is more prone to be triggered and leads to larger ventilation above the cloud layer, reducing stratocumulus clouds there. These two suites also have intrinsic differences in the interaction between cloud microphysics and other processes, resulting in different time step sensitivities. PhysC tends to generate more stratiform clouds with decreasing time step. This is caused by separate treatment of stratiform cloud condensation and other microphysical processes, leading to a tight interaction with boundary layer turbulence. In PhysW, all the microphysical processes are executed at the same temporal scale, and thus no such time step sensitivity was found.





## 1. Introduction

Global weather and climate modeling cover broad spatial and temporal scales. In their extreme manifestations, weather modeling is characterized by very high-resolution simulations (e.g., kilometer-level grid spacing), while climate modeling needs to deal with very long-term model integrations (e.g., multiple centuries). Weather forecasts are required to generate highly accurate and detailed atmospheric information within a relatively short period. The resultant model is typically designed to faithfully capture some process-level transient atmospheric features (e.g., extreme storms). In contrast, global climate modeling demands less biased mean climates with balanced energy and hydrological cycles. A realistic and stable model climate is typically the top priority to pursue, while those process-level weather details are of secondary interests. Such diverse application scenarios have led to significant differences in the formulations of weather and climate models. Among the model components, physics parametrization, which describes the unresolved (including under-resolved and non-resolvable) processes of an atmospheric model, tends to have more application-specific design choices across the scales of weather and climate modeling applications (e.g., Brown et al. 2012; Randall et al. 2018; Yu et al. 2019).

Global-to-Regional Integrated forecast SysTem (GRIST, version A22.7.28) is a unified model system for global weather and climate modeling (Li et al. 2022a; Li and Zhang 2022; Wang et al. 2019; Zhang et al. 2019; Zhang et al. 2020; Zhang et al. 2021; Zhou et al. 2020; Zhang et al. 2022). Currently, two major physics suites have been coupled to the dynamical core of GRIST. One suite (referred to as PhysC) is originally ported from a global climate model (Community Atmosphere Model, CAM, version 5.3). The other suite (referred to as PhysW) adopts several parameterization modules from a meso-scale weather model (Weather Research and Forecast, WRF, version 3.8.1). Previous studies have performed separate Atmospheric Model Intercomparison Project (AMIP) simulations based on GRIST-PhysW and GRIST-PhysC (Zhang et al. 2021; Li et al. 2022a). Zhang et al. (2021) showed that GRIST-PhysW produces a realistic model climate at relatively coarse resolutions (e.g., 120 km), with close-to-zero top-of-atmosphere (TOA) radiation budget. This is obtained by some manmade tuning of certain cloud physical properties and leads to a relatively large bias of net cloud radiative forcing (see Table 4 of Zhang et al. 2021). GRIST-PhysW can well simulate the global and regional precipitation patterns, especially a faithful replication of the diurnal cycles over East Asia, i.e., the contrasting regional features characterized by "afternoon" versus "nocturnal-to-early-morning" precipitation peaks.

In contrast, GRIST-PhysC can produce a nearly balanced TOA radiation budget with relatively smaller net cloud radiative forcing biases (see Table 3 of Li et al. 2022a). Model development experience





also suggests that GRIST-PhysC is more robust in terms of long-term simulation stability at the coarse resolution, while GRIST-PhysW needs to be more carefully configured to avoid potential long-term integration instability (e.g., the stability is sensitive to the choice of microphysics scheme). The simulated global and regional rainfall features of GRIST-PhysC, however, is overall inferior to that of GRIST-PhysW. For example, with increasing local resolution, GRIST-PhysW better simulates the eastward propagating rainfall episodes downstream of the Tibetan Plateau (Zhang et al. 2021), while GRIST-PhysC does not support such a beneficial resolution sensitivity even with a refined tuning of certain physical processes (Li et al. 2022b).

These discrepancies in the model physics suites raise interesting questions and motivate a further exploration of the different behaviors of the two physics suites. Because the AMIP experiments incorporate nonlinear dynamics-physics interaction and global-regional process feedback, it is rather important to understand the model behaviors in a more straightforward and isolated environment. This is achieved based on single column model (SCM) simulations in this study.

SCMs help to isolate the impact of the physics suite and evaluate its behavior in the absence of 3D dynamics (Zhang et al. 2016). It is commonly used for physical parameterization development and parameter tuning tests (Bogenschutz et al., 2012; Gettelman et al., 2019; Guo et al., 2014). It is also a computationally efficient tool to assess different schemes/models for specific physical processes and interactions such as tropical convection, cloud feedback, and diurnal variation of precipitation (Zhang et al. 2013; Davies et al. 2013; Tang et al. 2022). This study compares PhysC and PhysW and explores their key differences that are responsible for the contrasting model behaviors. Moreover, model sensitivity experiments are further performed to understand two specific model sensitivities related to PhysC and PhysW, as respectively found in the previous studies (see Sections 3.2&3.3). The general purpose of this study is to understand which processes and/or process chains have a dominate influence on the model performance and sensitivity.

The remainder of this paper is organized as follows. Section 2 briefly reviews the two physics suites and describes the single column model. The experimental design is given in section 3. Section 4 assesses the different behaviors that arise from the physical parametrizations and interprets some possible reasons responsible for these discrepancies. Section 5 explore two specific sensitivities related to the physics suites. Section 6 gives a summary.

## 2. Model description

### 2.1 The PhysC suite


The physical processes of PhysC are sequentially coupled with an order from the wet (deep and shallow convection, stratiform cloud condensation, and cloud microphysics) to dry (radiation transfer, surface flux, and planetary boundary layer, PBL turbulence) processes. It contains a mass-flux deep-convection parameterization scheme (Zhang and McFarlane 1995; ZM) with dilute convective available potential energy (CAPE; Neale et al. 2008) and modified convective momentum (Richter and Rasch 2008). An entraining–detraining bulk parameterization scheme is used for shallow convection (the University of Washington, UW scheme; Park and Bretherton 2009) with entrainment and detrainment rates determined by a buoyancy sorting algorithm (Kain and Fritsch 1990). The PBL turbulence is based on a downgradient diffusion of momentum and moist-conserved variables, with diffusivities calculated based on local turbulent kinetic energy (TKE) (Bretherton and Park 2009). The radiation transfer module is done by the Rapid Radiative Transfer Model for General circulation model (Iacono et al. 2008).

A fractional cloudiness condensation parameterization, together with a consistent diagnosed cloud fraction scheme, is separately evaluated in the model physics before the calculation of other microphysical process rates. This parameterization is referred to as cloud macrophysics in the context of CAM5's model physics (Park et al. 2014), which loosely inherits the more general concept of large-scale condensation (e.g., Rasch and Kristjansson 1998; Zhang et al. 2003). Large-scale condensation is conventionally used by global models that typically employ relatively coarse grid spacing. The sub-grid scale condensation of water vapor is treated via a Sundqvist-type scheme (Sundqvist 1978) with a prognostic treatment of stratus condensation and a diagnosed stratus cloud fraction. A grid box is separated into a cloudy and a clear-sky portion. The total cloud fraction is a sum of stratus fraction and cumulus fraction. The aerosol activation and microphysical processes then occur only within the cloudy portion of the grid box. Mathematically, this leads to a scaling of the microphysical process rates based on the cloud fraction. Cloud microphysics is calculated by a two-moment scheme that explicitly calculates the mass and number concentrations of cloud liquid and ice, rain, and snow (Gettelman et al. 2010; Morrison and Gettelman 2008), known as the MG scheme. Because large-scale condensation is an input for the following MG microphysics scheme, the underlying physical assumption is that the MG microphysics mainly deal with cloud dynamics related to stratiform-like clouds, irrelevant of how cloud fraction is defined.

## 2.2 The PhysW suite

In the 3D model, PhysW is coupled to the GRIST dynamical core in a different way from PhysC. In the SCM, however, because there is no two-way dynamics-physics interaction, a sequential approach





similar to PhysC is adopted for coupling the physics schemes. Cloud microphysics (WSM6; Hong and Lim 2006) is computed first, followed by surface flux computation. WSM6 generates microphysical process rates for six species (water vapor, cloud liquid and ice, rain, snow, and graupel) and the associated potential temperature tendency. The sedimentation of falling hydrometeors is computed before other

microphysical processes, which is different from the MG scheme that computes the "microphysics" first. Condensation from water vapor to cloud water is calculated after all other microphysical processes, only when the entire grid box is supersaturated (Yao and Austin 1979). When coupled to the dynamical core, PhysW has a clear difference from PhysC, that is, dynamics and all the microphysical processes are more closely coupled together, and microphysics is not specifically tied to those physical assumptions related

to large-scale stratiform-like clouds. It ensures a more natural transition of this model formulation to global "storm-resolving" setup as the resolution is refined.

PBL turbulence, cumulus convection, and radiation transfer are sequentially called after the atmospheric state updated by the cloud microphysics scheme. The Yonsei University (YSU) scheme based on the non-local-$K$ approach is used for PBL turbulence (Hong and Pan, 1996). A modified Tiedtke-

Bechtold (TB) convective scheme from European Center for Medium-range Weather Forecast is used to calculate deep-, shallow-, and middle-level convection (Zhang and Wang 2017). Deep and shallow convection share the same cloud function, while use different trigger-closure assumptions and entrainment-detrainment rates. They do not co-occur within one time step. The detrained cloud condensates are returned to the grid-scale cloud liquid/ice following a probability function dependent on

temperature. The shortwave and longwave radiation transfer of PhysW uses the RRTMG, although the code is somewhat different from that of PhysC. Cloud fraction is purely diagnosed. In this study, we use the Xu and Randall (1996) scheme. It is based on the cloud condensate and snow at the time slice before the radiation transfer.

### 2.3 The single column model

In addition to the software aspect of handling integration workflow, data diagnostics and I/O, the main part of the GRIST single column model contains a simplified dynamical core to handle the vertical advective processes of temperature ($T$) and water vapor ($q_v$) within an atmospheric column.

$$\frac{\partial T}{\partial t} = -(\vec{V} \cdot \nabla T)_{LS} - \omega_{LS}\frac{\partial T}{\partial p} + \frac{R_d T}{p c_p}\frac{dp}{dt} + \left(\frac{\partial T}{\partial t}\right)_{phys} + [\frac{(T - T_{obs})}{\tau}]_{rex}, \qquad (1)$$

$$\frac{\partial q_v}{\partial t} = -(\vec{V} \cdot \nabla q_v)_{LS} - \omega_{LS}\frac{\partial q_v}{\partial p} + \left(\frac{\partial q_v}{\partial t}\right)_{phys} + [\frac{(q_v - q_{v_{obs}})}{\tau}]_{rex}, \qquad (2)$$

where $p$ and $\omega$ are pressure and pressure-based vertical velocity; $R_d$ represents gas constant for dry



air, and $c_p$ the heat capacity at constant pressure for dry air; subscript *phy* denotes the physical parameterizations, *LS* stands for the large-scale fields, and *obs* stands for observed values. Here, the *rex* term represents relaxation with the time scale $\tau$. The SCM predicts temperature and humidity using the prescribed large-scale horizontal tendencies as forcing terms, together with the subgrid-scale tendencies

provided by the physical parameterization. A two-time-level predictor-corrector time integrator (Wicker and Skamarock 2002) is used. The approximation of $T$ and $q_v$ values at the interface levels follows a standard line-based third-order upwind flux operator:

$$q_{k+\frac{1}{2}} = \frac{7}{12}(q_{k+1} + q_k) - \frac{1}{12}(q_{k+2} + q_{k-1}) + sign(\omega_{k+\frac{1}{2}})\frac{1}{12}[(q_{k+2} - q_{k-1}) - 3(q_{k+1} - q_k)]. \qquad (3)$$

Eq. (3) gives the approximation of $q_v$ as an example, in which subscript *k* represents vertical layer index,

and *k*+1/2 stands for the interface level. The momentum, pressure-based vertical velocity, and surface pressure at each integration step are provided by the Intensive Observation Period (IOP) dataset.

## 3. Experimental design

### 3.1 Field cases for performance comparison

Three SCM field cases over the ocean are selected to assess the two physics suites (Table 1). The

Tropical Warm Pool International Cloud Experiment (TWP-ICE) is widely used to study the representation of rainfall and cloud associated with tropical convection. The second Dynamics and Chemistry of Marine Stratocumulus Experiment (DYCOMS) focuses on the nonprecipitating marine stratocumulus clouds. In addition to two short-term process-oriented studies, a long-term simulation (the CFMIP-GASS Intercomparison of LES and SCM experiment; CGILS) is further conducted to investigate

the statistics of cloud and its radiative forcing. The two physics configurations use the same vertical resolution (30 full layers with a top at 2.25 hPa). The time step (*dt*) for physical processes is 1200 s except for the radiation transfer (*dt_rad* = 3600 s). During the time steps when the radiation transfer model is not active, the previously saved tendencies are used to update the atmospheric state.

It is useful to use the moisture budget equation to probe the key physical interactions responsible

for the diverse behaviors in the SCM. The sum of the physical tendencies in this direct approach corresponds to the "observed" apparent drying ($Q_2$; Yanai et al. 1973) for estimating the bulk effect of diabatic processes. Following Zhang et al. (2013), the water vapor budget can be written as:

$$\frac{\partial q_v}{\partial t} = \left(\frac{\partial q_v}{\partial t}\right)_{PBL\_turb} + \left(\frac{\partial q_v}{\partial t}\right)_{conv} - (c - e)_{microp} - \left[\left(\overline{V \cdot \nabla q}\right)_{LS} + \omega_{LS}\frac{\partial q_v}{\partial p}\right], \qquad (4)$$

containing the large-scale forcing (*LS*) and three physical parameterization terms, i.e., PBL turbulence



(*PBL_turb*), convection (*conv*), and large-scale net condensation by microphysics (*c-e*)$_{microp}$. For PhysC, the microphysical condensation term represents the sum of macrophysics and MG microphysics, and the convection term contains ZM deep and UW shallow convection.

### 3.2 Simulations with and without parameterized convection

In addition to the baseline simulations from different SCM field cases, two additional groups of
sensitivity experiments were further performed. These two sensitivity experiments intend to closely answer the questions raised in the earlier 3D model simulations using the two physics suites, respectively. The first group turns off the convective parameterization and compares the simulated precipitation and clouds with those generated by the parameterized convection runs. As demonstrated by Zhang et al. (2022), the direct dynamics-microphysics interaction of GRIST-PhysW tends to produce artificially
abundant tropical cloud liquid water mixing ratio and precipitation rates in the absence of parameterized convection, especially when the grid spacing is coarser than the so-called "storm-resolving" scale. In this study, we use a more isolated environment to demonstrate that such a result is closely related to a direct response of the microphysical processes when forced by large-scale advective forcing. We also compared the behaviors of PhysW and PhysC under this setup.

### 3.3 Sensitivity of the physics suites to time step

The second sensitivity experiment assessed the time step sensitivity due to the different process coupling and/or parameterizations of fast processes. Previous studies based on the CAM-family model physics all demonstrated a relatively strong sensitivity to the time step (e.g., Williamson 2013; Wan et al. 2015; Li et al. 2020; Santos et al. 2021). Wan et al. (2015) suggested that the representation of stratiform
cloud processes in CAM5 largely reduced the time step convergence rate in the short-time time step convergence test. Li et al. (2020) also found a clear time step sensitivity of CAM5 in the tropical cyclone simulations, and they noted the grid-scale condensation increased with the shortening time step and the more frequent coupling to dynamics, which enhanced the tropical cyclone and precipitation.

In this study, we explore a possible physical mechanism responsible for the time step sensitivity
and compare the behaviors between the two physics suites. The time step for the physical processes varies among the 2400s, 1200s, 600s, 300s, and 100s except for radiation transfer. The radiative heating varies relatively slowly and thus has a very small impact on the model sensitivity to time step (Santos et al. 2021; Wan et al. 2021).



## 4. Intercomparison of simulation performance

### 4.1 Tropical convection: TWP-ICE

The TWP-ICE is divided into a convection active period for the first 6 simulation days and a relatively suppressed period thereafter (Davies et al. 2013). The simulated precipitation rates for the two physics suites are mutually consistent and overall close to the observation during the convection active period. The simulated peak values at day 5 are about 50 mm day$^{-1}$ smaller than the observed value (Figure 1a). A notable discrepancy is found in the precipitation partitioning, where the ratio between convective and total precipitation rates in PhysW is larger than that of PhysC. The convective rainfall dominates the total rainfall. Meanwhile, PhysW better captures the onset and retreat of rainfall events during both periods, while PhysC tends to produce artificially weak rainfall in the intervals of major rainfall peaks (Figures 1a and 1b). This implies that when large-scale forcing is not strong enough to generate strong convective events, the ZM scheme of PhysC is more prone to be triggered by weak local forcing, as compared with the TB scheme in PhysW.

Different trigger-closure assumptions of the two convection schemes can largely explain the simulated precipitation differences. The TB scheme adopts a dynamic-like convective equilibrium (Bechtold et al. 2014; Zhang and Wang 2017). A "first-guess" updraught depending on a mixed layer (i.e., an average of the lowest 60 hPa) is adopted to determine the cloud base height (i.e., the lifting condensation level) and cumulus properties at the cloud base. Such a deep source layer requires sufficient mixing by grid-scale dynamics (and/or sub-grid scale turbulence) and avoids spurious weak convection. Deep convection occurs only when the cloud base is found, and the cumulus cloud thickness can be thicker than 200 hPa. In PhysC, the ZM deep convection is triggered when the dilute CAPE is greater than 70 J/kg. The strength of convection is determined by a fixed consumption rate of CAPE. This design feature tends to more frequently trigger deep convection than that in the TB scheme, especially during the convection suppressed period with weak large-scale forcing.

Figures 1 (c-f) compare the period-averaged cloud fraction and cloud liquid water mixing ratio between PhysW and PhysC. The shape of cloud profiles for PhysC overall resembles the observation from IOP. It overestimates ~0.2 middle and upper-level cloud fraction (200-600 hPa) for the convection active period and produces ~0.15 larger low-level cloud fraction (700-900 hPa) for the suppressed period. In contrast, PhysW shows a notable underestimation of low cloud fraction in the convection active period. By analyzing the water vapor budget, it is found that vertical transport and/or condensation of water vapor by the TB scheme of PhysW are stronger than the ZM scheme of PhysC (Figure 2a). It implies that parameterized convection in PhysW plays a more important role in water vapor transport and rainfall



formation than that in PhysC. The stronger vertical moisture transport by the TB deep convection reduces the low-level cloud liquid mixing ratio and the corresponding diagnosed cloud fraction. PhysW also underestimates low clouds in other tropical convection cases such as the GATE Phase III (figure omitted).

For these two suites, the different treatments of dynamics-microphysics interaction may also explain the differences in the simulated cloud profiles. This can be studied based on the convection suppressed period of TWP-ICE (Figure 1d), in which dynamics-microphysics interaction plays a more significant role than the parameterized convection. For PhysC, the fractional cloudiness condensation is prognosed following a triangular probability density function. The cloud fraction is diagnosed based on the prognostic cloud condensate before calling other microphysical processes. Cloud condensate is a direct

source for other microphysical processes. The MG microphysics consumes cloud water but does not alter the cloud fraction. This corresponds to the "emptier low cloud" feature of PhysC as compared with PhysW (Figures 1c-f), that is, larger cloud fraction with lower cloud liquid content.

     For PhysW, cloud condensation is handled as part of the explicit microphysics-dynamics coupling. Condensation is computed at the final stage of WSM6. If supersaturation is detected after all other

microphysical processes, cloud condensate will be generated; otherwise, clouds evaporate. Cloud fraction is diagnosed based on the cloud condensate and snow mixing ratio after the convection and microphysics processes. Therefore, it produces a smaller cloud fraction below 600 hPa than PhysC because the grid-scale mean state is more difficult to reach supersaturation after the convective and microphysical precipitation processes, especially for relatively large grid intervals.

**4.2 Marine stratocumulus cloud (DYCOMS) and stratus to shallow convection transition (CGILS)**

     These two cases specifically focus on the stratiform-like clouds, which is a major source of cloud water that exerts a large influence on the shortwave cloud radiative forcing. The DYCOMS is an idealized test case of steady nocturnal stratocumulus under a dry inversion with embedded pockets of drizzling open cellular convection (Figure 3a). The time-averaged cloud properties for PhysW and PhysC are in

good agreement (Figures 3b and 3c). The modeled low-level cloud fractions are concentrated within a layer between ~900 and 950 hPa, and the maximum value reaches one at 920 hPa. The cloud in PhysC is thicker than that in PhysW. Despite the consistent stratus amount, the interactions of the physical processes to generate clouds are different between PhysW and PhysC (Figure 3d). In PhysC, the PBL turbulence moistens the lower levels, and the macrophysics condenses water vapor to generate clouds.

In PhysW, the shallow convection is active for transporting moisture in addition to the PBL turbulence. The collaborative effect of shallow convection and PBL turbulence in PhysW is weaker than that of the PBL turbulence in PhysC. Cloud condensation in the WSM6 microphysics is also weaker than the





macrophysics of PhysC.

CGILS is a long-term integration experiment to investigate the statistics for cloud fields. It simulates
the cloud transition from coastal stratus to shallow cumulus offshore along the Pacific Cross-Section
Intercomparison region in the north tropical to subtropical Pacific (see Figure 4 in Zhang et al. 2013).
Three locations are selected to model different regimes of clouds, i.e., shallow cumulus at CGILS-S6,
stratocumulus at CGILS-S11, and well-mixed stratocumulus or coastal stratus CGILS-S12 (Table 1).
Both PhysC and PhysW reach quasi-equilibrium after a few days. They overall reproduce the transition
characteristics from stratus at CGILS-S12 to shallow cumulus at CGILS-S6, that is, cloud top and cloud
thickness increase and cloud fraction decreases (Figure 4). PhysC resembles the cloud radiative forcing
at CGILS-S6, but underestimates it at CGILS-S11 and CGILS-S12 (Table 2). PhysW has ~0.4 larger low
cloud fraction than PhysC at CGILS-S12, and it generates a notably stronger cloud radiative forcing at
this location. PhysW has a sharper decline across the transitions from CGILS-S12 to CGILS -S11, thus
it substantially underestimates the cloud radiative forcing at CGILS-S11. It implies an earlier occurrence
of stratocumulus-to-cumulus transition in PhysW. At CGILS-S6, shallow convection of PhysW is less
frequently triggered than that in PhysC and produces higher and slightly larger shallow cumulus clouds.

The water vapor budget shows that more active shallow convection in PhysW is the major
contributor to the discrepancy in cloud transition (Figure 5). In PhysW, the shallow convection is active
at CGILS-S12 to transport moisture upward in addition to the turbulence. The collaborative effect of
shallow convection and PBL turbulence in PhysW plays a similar role in moisture transport as the PBL
turbulence in PhysC. Condensation produced by WSM6 of PhysW is greater than that from fractional
condensation parameterization of PhysC, facilitating the generation of cloud. At CGILS-S11, the active
shallow convection of PhysW causes a stronger ventilation effect above the cloud layer, thus evaporation
can occur in the WSM6 microphysics, reducing the low cloud. In PhysC, in addition to the grid-scale
dynamical advection, only two physical mechanisms are active to produce status at CGILS-S11 and
CGILS-S12, that is, turbulence moistens the PBL layers, and the fractional cloud condensation dries it.

## 5. Intercomparison of simulation sensitivity

### 5.1 Cloud and precipitation simulations in the absence of parameterized convection

The base experiments suggest that the convective parameterization is a major source of uncertainty
in the SCM simulated clouds and precipitation. As mentioned in Section 3.2, a group of sensitivity
experiments that turned off the convective parameterization was further performed. We note that unlike



in the previous 3D simulations in Zhang et al. (2022), the SCM only supports one-way feedback from dynamics to microphysics. The prescribed large-scale advective tendencies do not respond to the microphysical process rates. The TWP-ICE and CGILS cases are used to reveal the responses of tropical precipitation and clouds in the absence of parameterized convection. We also assess the differences between the two physics suites under this setup. The tests without parameterized convection are referred to as "nocu".

As shown in Figures 6a and 6b, in the TWP-ICE case, the "nocu" runs of PhysC and PhysW produce highly consistent precipitation evolution, especially during the convection active period. This implies that precipitation generated by the cloud microphysical processes in response to a strong large-scale forcing is consistent across the two physics suites. The smaller difference in the water vapor budgets between PhysC and PhysW supports this argument (comparing Figure 2 and Figures 6e, 6h). This also confirms that for precipitation simulations, the convective parameterization is the primary source of model discrepancy when it is active.

Figures 6 (c, d, f, and g) show that the microphysics-dynamics coupling of both PhysC and PhysW produces more abundant cloud liquid water and cloud fraction as compared with those in the base simulations with the active convective parameterization. Increase of the middle and low clouds (500-900 hPa) is more notable for PhysW. This is in accordance with the 3D global simulation with explicit dynamics–microphysics coupling (Zhang et al. 2022). The mechanism responsible for the changes in middle and low clouds can be studied by comparing the water vapor budgets in Figure 2a and Figure 6e. Deep convective parameterization is designed to represent penetrative under-resolved scale vertical motions, including sub-grid scale eddy transport of heat, moisture, and momentum. Middle- and low-levels are stabilized and unsaturated because of convection (stratiform cloud evaporation is found in Figure 2a), leading to a relatively small cloud fraction. For the simulations without parameterized convection, cloud microphysics can directly respond to the grid-scale destabilization, e.g., via condensational drying. Therefore, the direct interaction between grid-scale motion and microphysics tends to generate overly large cloud liquid mixing ratio and low cloud fraction. The substantial middle- and low-level cloud condensate quantities associated with microphysics than that associated with convection was also noted in other models such as GFDL-GFS (Lin et al., 2013).

The more abundant cloud liquid water and larger and lower cloud fraction are also found in the "nocu" runs of CGILS, especially for PhysW at CGILS-S6 and CGILS-S11 (Figure 7). The maximum cloud fraction in the "nocu" run for PhysW reaches one at CGILS-S6 and CGILS-S11, and the maximum cloud liquid mixing ratio is nearly 4 times larger than that in its base run. This corresponds to enhanced cloud radiative forcing, which becomes notably larger than the observation at the two locations (Table



2). PhysC shows changes only at CGILS-S6, where cloud develops slightly higher and the maximum cloud fraction increases ~0.2. Figure 8 shows that in the absence of parameterized convection, the water vapor tendencies of both PBL turbulence and microphysics increase to balance the budget in PhysW. This interaction of PBL turbulence and microphysics to generate stratiform clouds at CGILS-S11 and

CGILS-S12 is similar to that in PhysC, but the cloud condensate by microphysics is much larger. The enhanced microphysical condensation thus increases the low cloud fraction and cloud liquid water. This also highlights the role of convective parameterization in the vertical transport of heat and moisture for cloud generation in PhysW.

### 5.2 Sensitivities of physical interactions to time step

Previous studies using the CAM-family model physics suggested that time step size has significant effects on the simulated precipitation and clouds. Wan et al. (2015) suggested that the fractional cloudiness condensation was the major contributor to the time step sensitivity. The fractional cloudiness condensation is widely used by global climate models because of the relatively coarse grid spacing, while in PhysW, instantaneous condensation is executed with other microphysical processes at the same

temporal scale. In this section, we use the CGILS case to compare the time step sensitivities related to the cloud process between PhysC and PhysW.

Figure 9 shows the time-averaged cloud fraction and cloud liquid mixing ratio using different time step sizes. It is seen that PhysW and PhysC show sensitivities to time step at different locations. The cloud property for PhysW with $dt = 2400$ s is largely different from other $dt$ runs, implying an

abnormal model performance caused by the overly long time step. Apart from $dt$=2400 s, the cloud and cloud radiative forcing are overall insensitive when varying the time step at CGILS-S11 and CGILS-S12, but they show sensitivity to the time step at CGILS-S6. The cloud fraction and cloud liquid mixing ratio for $dt$=1200 s are 0.3 and 0.09 g/kg, respectively, and they decrease with shortening time step, reducing the cloud radiative forcing over this location (Figures 9a and 9d, and Table 2). The shallow convective

mass fluxes for different time step sizes demonstrate that the shallow convection slightly weakens with decreasing time step, reducing the source of cloud water (Figure 10a).

The PhysC simulated stratiform cloud at CGILS-S11 and CGILS-S12 is more sensitive to the time step size than PhysW (Figures 9 g-l). The maximum cloud fraction at CGILS-S12 is about 0.4 for $dt$= 2400 s, and it increases by more than a factor of 2 when the time step is shortened to 300 and 100 s. The

cloud liquid water also shows an increase with the decreasing time step, enhancing the cloud radiative forcing (Table 2). The positive feedback between the macrophysics and PBL turbulence can explain the sensitivity of the stratiform cloud to time step (Figure 10b). At CGILS-S12, the stratiform condensation





of the macrophysics activates in response to the moistening by PBL turbulence. The water vapor tendencies for macrophysics and turbulence increase with the decreasing time step. It implies that the

increased stratiform condensation in the shorter time step run enhances the vertical downgradient diffusion of moisture by PBL turbulence, which in return generates more stratiform condensation that dries the atmosphere. Wan et al. (2014) also found that the cloud fraction in CAM5, accompanied by the ice and liquid water path, increases with the decreasing time step, especially over the trade wind regions. This is a numerical issue associated with compensating processes that can be significantly sensitive to

time step (Wan et al. 2013).

### 6. Summary

This study makes an intercomparison of the weather (PhysW) and climate (PhysC) physics suites in a unified forecast/climate model system (GRIST-A22.7.28) using SCM simulations. The discrepancy of simulated precipitation and cloud fields due to different physics suites was studied. The major

conclusions are summarized as follows.

The SCM simulations demonstrate that the convective parameterization contributes to the major discrepancy of precipitation and clouds between the two suites. The Tiedtke-Bechtold convective parameterization of PhysW better captures the onset and retreat of rainfall events than the Zhang&McFarlane scheme of PhysC. Meanwhile, the stronger vertical moisture transport by convection

leads to an underestimation of the middle- and low-level cloud fraction for PhysW. Over the typical stratus-to-stratocumulus transition regime such as the Californian coast, shallow convection of PhysW is more prone to be triggered. The collaborative effect of shallow convection and PBL turbulence in PhysW provides a similar effect for moisture transport as the PBL turbulence in PhysC. Meanwhile, the more easily triggered shallow convection in PhysW can reduce low clouds over the cloud transition regions

because of the larger ventilation above the cloud layer. When switching off the convective parameterization, the precipitation formation by microphysics in response to the large-scale forcings is consistent across the two physics suites. Both PhysC and PhysW will produce more abundant cloud liquid water and low cloud fraction if the bulk effects of vertical transport of moisture and heating by parameterized convection are absent.

The interaction between microphysics and other processes also explains the discrepancy of simulated low clouds between PhysW and PhysC. The grid-scale condensation (evaporation) of PhysW is addressed as one of the microphysical processes in the WSM6 scheme. It is calculated lastly if grid-scale supersaturation (unsaturation) occurs after all other microphysical processes. The cloud fraction is



diagnosed after the microphysical and convective processes. In contrast, PhysC prognoses stratiform cloud condensation and diagnoses cloud fraction before other microphysical processes. The cloud condensate is the direct source of microphysics. Therefore, PhysC tends to produce a larger low cloud fraction than PhysW. This separate treatment of fractional cloudiness condensation and other microphysics processes can cause a tight interaction with boundary layer turbulence, leading to sensitivity to time step size in simulating stratiform clouds. In the absence of fractional cloudiness

condensation in PhysW, the assumption that condensation of water vapor occurs at the same temporal scale with other microphysical processes does not show such time step sensitivity.

The intercomparison via SCM provides a guidance for understanding the model physics suites. PhysW has a higher skill to capture rainfall events, but the underestimated low clouds need to be ameliorated, because it is important to the energy budget. PhysC has a more sophisticated representation

of stratiform cloud condensation, cloud fraction and other microphysical processes, thus producing more reasonable cloud fields. However, too frequent convection deteriorates the simulation of precipitation. A more proper representation of parameterized convection, cloud condensation, microphysics and their interactions with model dynamics is important for achieving a unified model physics suite for future model development.

**Code and data availability.** A frozen version of the model code for supporting this manuscript and the model output data are available at: https://doi.org/10.5281/zenodo.7350131.

**Author contributions.** X. Li developed the single column model, coupled the PhysC suite, and conducted the SCM experiments. Y. Zhang coupled the PhysW suite and maintained the workflow of GRIST-A22.7.28, with contributions from X. Peng and J. Li. X. Li and Y. Zhang provided materials and

contents for this manuscript with contributions from B. Zhou and Y. Wang. All the authors continuously discussed the model development and the results of this manuscript.

**Competing interests.** The authors declare that they have no conflict of interest.

**Acknowledgement.** This study is supported by the National Natural Science Foundation of China (42205160 and 41875135).

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





Table 1. A list of single column model test cases

| Case | Long name | Lat, Lon | Type | Date | Length | Reference |
|------|-----------|----------|------|------|--------|-----------|
| TWP-ICE | Tropical Warm Pool International Cloud Experiment | (-12, 131) | Tropical convection | Jan 2006 | 14 days | (Davies et al. 2013) |
| DYCOMS | Dynamics of Marine Stratocumulus Experiment | (32, 239) | Stratocumulus | Jul 2001 | 1 day | (Stevens et al. 2005) |
| CGILS-S6 | CFMIP-GASS Intercomparison of LES and SCM | (17, 211) | Shallow cumulus | Jul 1997 | 150 days | (Zhang et al. 2013) |
| CGILS-S11 | CFMIP-GASS Intercomparison of LES and SCM | (32, 231) | Stratocumulus | Jul 1997 | 150 days | (Zhang et al. 2013) |
| CGILS-S12 | CFMIP-GASS Intercomparison of LES and SCM | (35, 235) | Stratus | Jul 1997 | 150 days | (Zhang et al. 2013) |






**Table 2.** Cloud radiative forcing of CGILS for PhysW and PhysC (unit: W m$^{-2}$)

| Name | CGILS-S6 (OBS: -23.4) | | CGILS-S11 (OBS: -82.57) | | CGILS-S12 (OBS: -84.35) | |
|---|---|---|---|---|---|---|
| | PhysW | PhysC | PhysW | PhysC | PhysW | PhysC |
| dt=2400 s | -100.39 | -32.87 | -38.25 | -28.97 | -10.55 | -17.07 |
| dt=1200 s | -54.81 | -28.24 | -7.68 | -46.79 | -103.48 | -31.91 |
| dt=600 s | -17.40 | -32.22 | -3.69 | -58.75 | -121.51 | -52.10 |
| dt=300 s | -7.04 | -29.38 | -1.16 | -60.59 | -125.37 | -67.78 |
| dt=100 s | -0.93 | -28.59 | -0.02 | -59.57 | -126.45 | -71.81 |
| nocu | -141.99 | -24.41 | -127.34 | -46.79 | -126.26 | -31.91 |

Note: The base run and "nocu" run use a default time step ($dt$=1200 s). OBS represents the observation

from JJA mean of CERES-EBAF dataset (Loeb et al. 2009).





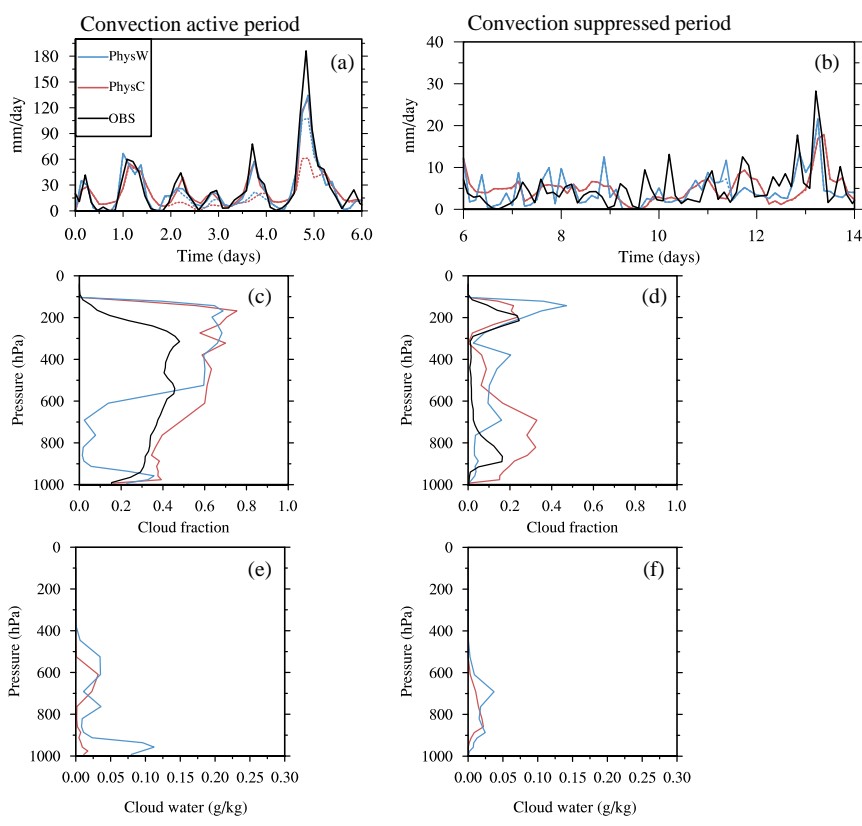


**Figure 1:** Time series of precipitation (solid, units: mm day$^{-1}$) and convective precipitation rates (dashed) for the (a) convection active and (b) suppressed periods of TWP-ICE. Shown are PhysW (blue), PhysC (red), and the IOP observation (black). (c and d) Time-averaged cloud fraction and (e and f) cloud liquid water mixing ratio (units: g kg$^{-1}$) for the two periods of TWP-ICE.


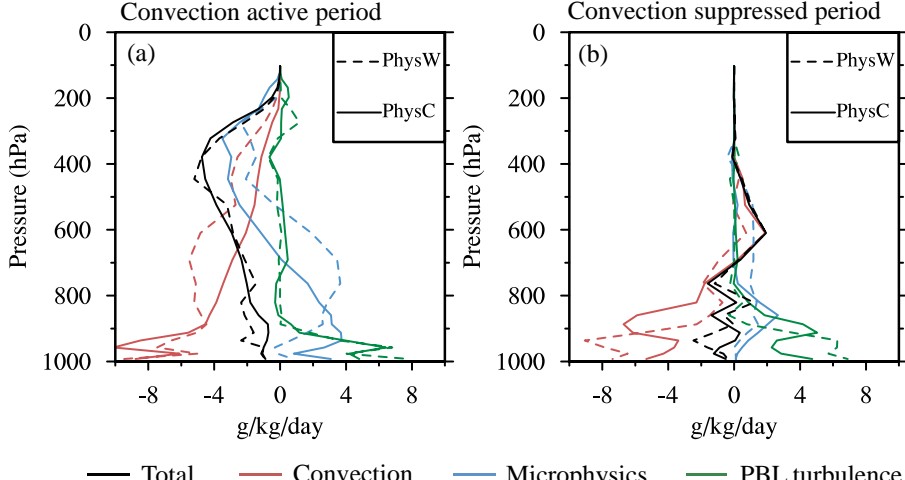

**Figure 2:** Time-averaged water vapor budget simulated by PhysC (solid lines) and PhysW (dashed lines) for the (a) convection active and (b) suppressed periods of TWP-ICE (units: g kg$^{-1}$ day$^{-1}$). Shown are the net water vapor tendency (black) and the effect of convection (red), large-scale stratiform net condensation (microphysics, blue), and PBL turbulence (green). For PhysC, the convection is represented by the sum of deep and shallow convection, and the large-scale stratiform net condensation contains the effect of both macrophysics and microphysics.





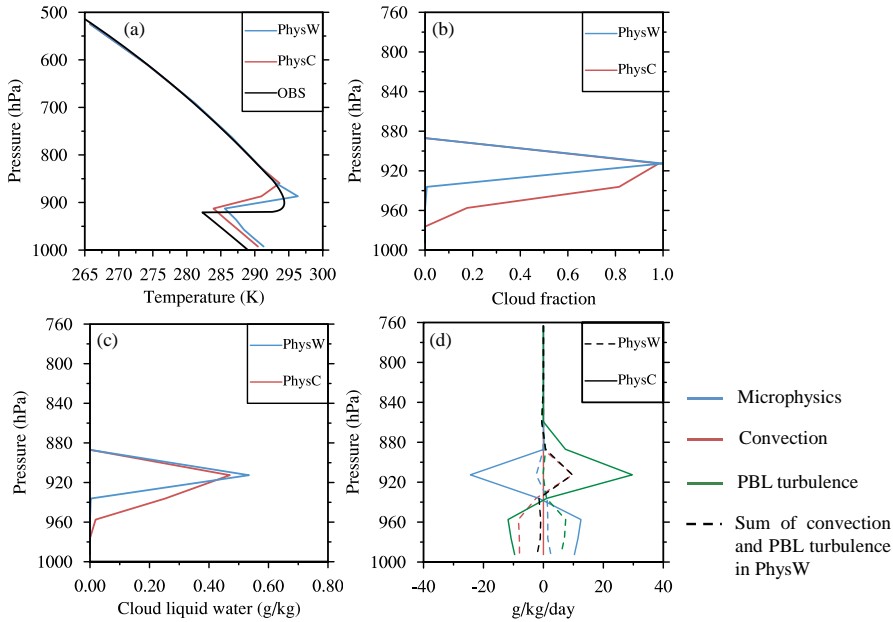

**Figure 3:** Time-averaged (a) temperature (units: K), (b) cloud fraction, and (c) cloud liquid water mixing
ratio (units: g kg$^{-1}$) simulated by PhysC (red) and PhysW (blue) for DYCOMS. The black solid line in
(a) shows the observed temperature. (d) Time-averaged water vapor budget (units: g kg$^{-1}$ day$^{-1}$) for PhysC
(solid lines) and PhysW (dashed lines). Shown are water vapor tendencies of large-scale stratiform net
condensation (microphysics, blue), shallow convection (red), and PBL turbulence (green). The black
dashed line represents the sum effect of shallow convection and PBL turbulence in PhysW.





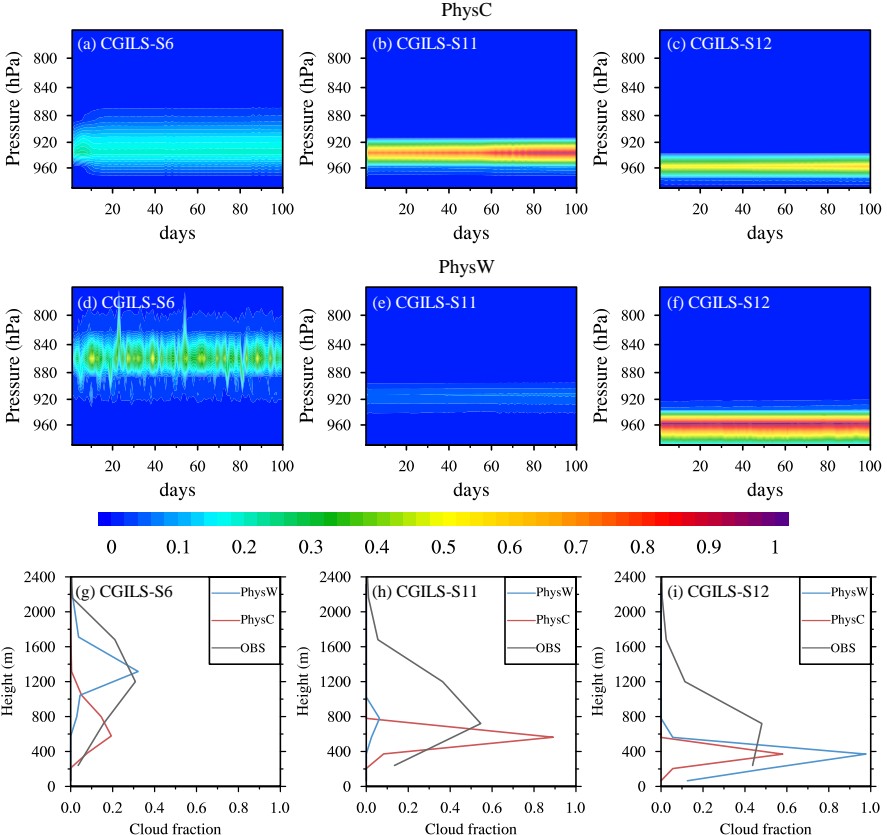

**Figure 4:** Time-pressure cross sections of cloud fraction simulated by PhysC for (a) CGILS-S6, (b) CGILS-S11, and (c) CGILS-S12. (d-f) The same as (a-c) but from PhysW. (g-i) Comparison of the time-averaged cloud fraction simulated by PhysC (red) and PhysW (blue) with the CALIPSO GOCCP data set (OBS, gray, Chepfer et al. 2010) at (g) CGILS-S6, (h) CGILS-S11, and (i) CGILS-S12. It is noted that the CALIPSO GOCCP data sensed by lidar may underestimate low stratus because the optically thick clouds will attenuate the lidar signal.






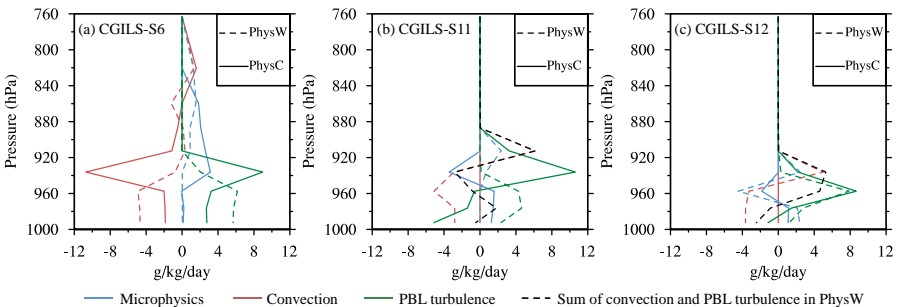

**Figure 5:** Water vapor budget for PhysC (solid) and PhysW (dashed) at (a) CGILS-S6, (b) CGILS -S11, and (c) CGILS-S12 (units: g kg$^{-1}$ day$^{-1}$). The color indexes for the water vapor tendencies follow that in Figure 3 (d).





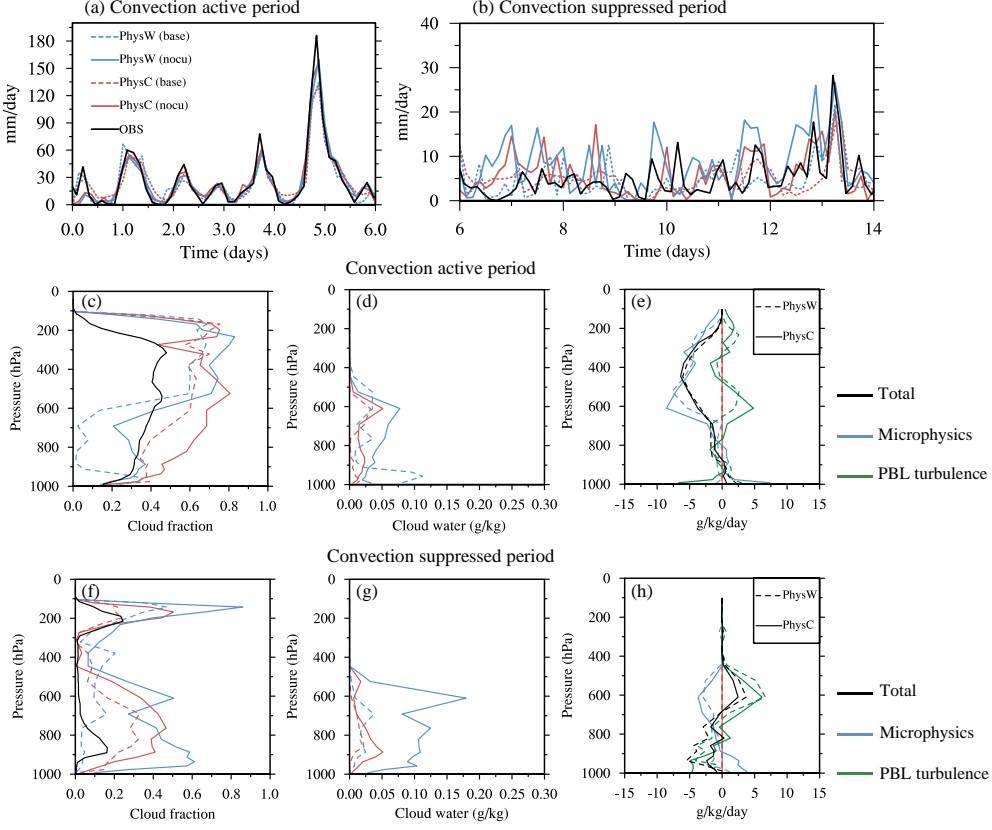

**Figure 6:** Time series of precipitation (units: mm day$^{-1}$) for the (a) convection active and (b) suppressed periods of TWP-ICE, and time-averaged (c) cloud fraction and (d) cloud liquid water mixing ratio (units: g kg$^{-1}$) for the convection active period. Shown are the observation from IOP (black solid lines) and simulations for PhysC (red solid lines) and PhysW (blue solid lines) without parameterized convection (nocu). The base runs using parameterized convection (same as that in Figure 1) are also illustrated for comparison (dashed lines). (f and g) The same as (c and d) but for the convection suppressed period of TWP-ICE. The time-averaged water vapor budget for (e) the convection active and (h) suppressed periods simulated by PhysC (solid lines) and PhysW (dashed lines) in the absence of parameterized convection. The color indexes for the water vapor tendencies follow that in Figure 2.



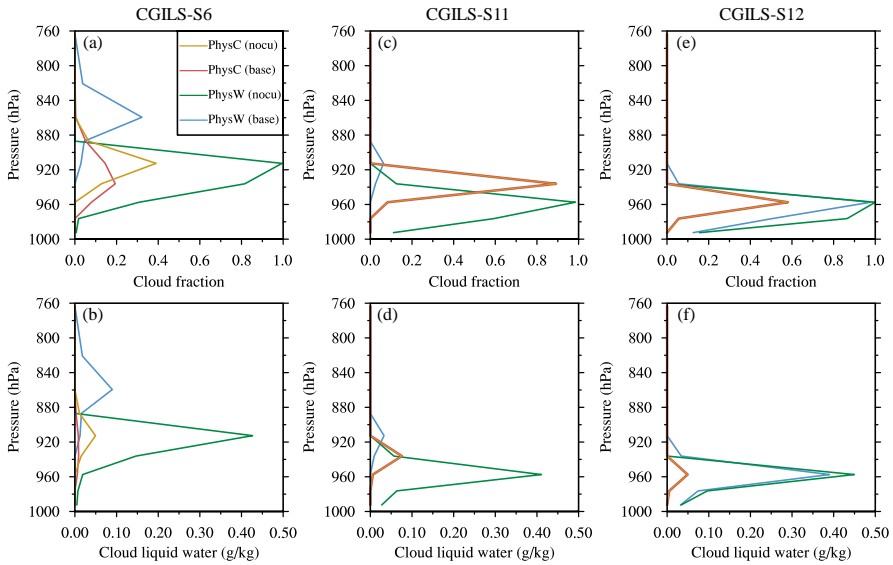

**Figure 7:** Time-averaged (a) cloud fraction and (b) cloud liquid water mixing ratio (units: g kg$^{-1}$) for CGILS-S6. Shown are the simulations for PhysW without the parameterized convection ("nocu", green) and its base run (blue), and the PhysC "nocu" (yellow) and the base runs (red). (c-d) and (e-f) The same as (a-b) but for CGILS-S11 and CGILS -S12.



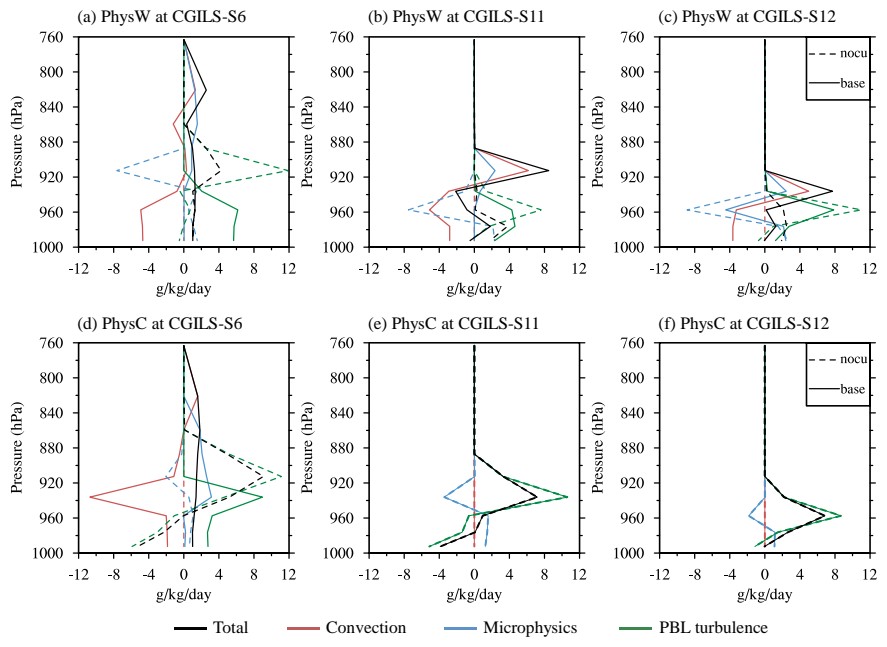

650

**Figure 8:** Comparison of the water vapor budget (units: g kg⁻¹ day⁻¹) between the base run (solid) and that without the convective parameterization ("nocu", dashed) for PhysW for (a) CGILS-S6, (b) CGILS-S11, and (c) CGILS-S12. The color indexes for the water vapor tendencies follow that in Figure 2. (d-f) The same as (a-c) but for PhysC.



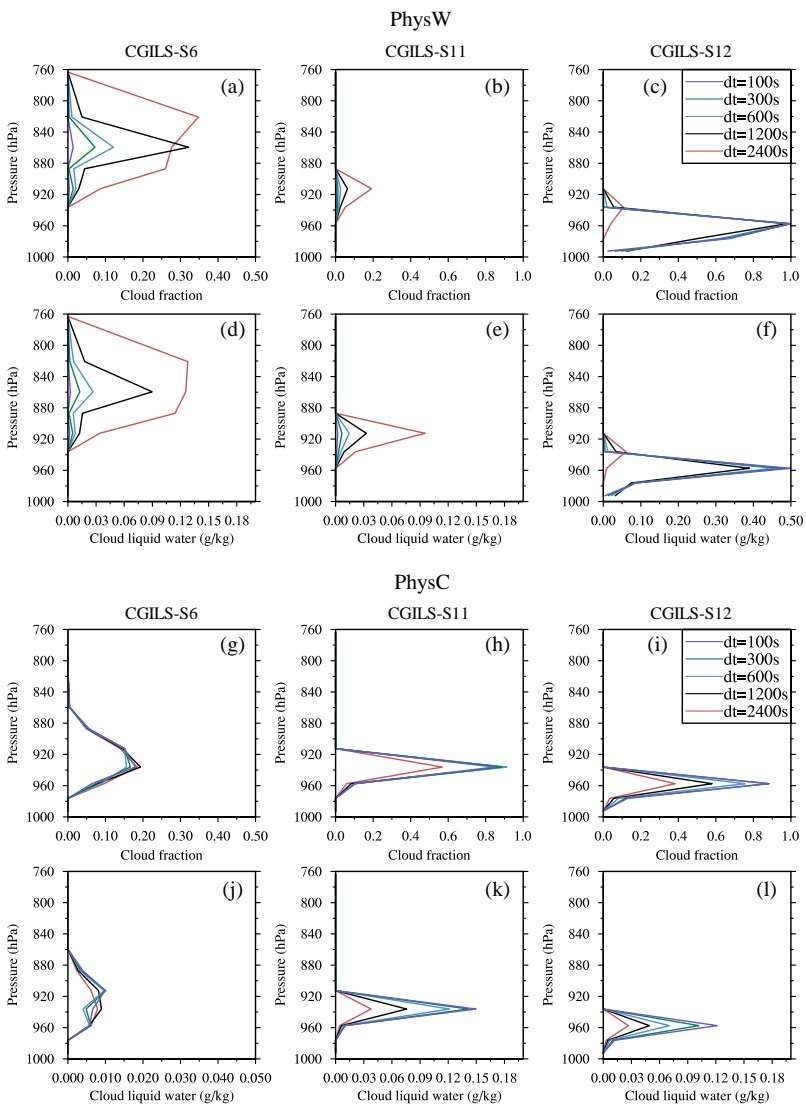

**Figure 9:** Time-averaged cloud fraction for (a) CGILS-S6, (b) CGILS-S11, and (c) CGILS-S12 modeled by PhysW with *dt*=2400s, 1200s, 600s, 300s, and 100s. (d-f) The same as (a-c) but shows the time-averaged cloud liquid water mixing ratio (units: g kg$^{-1}$). (g-l) The same as (a-f) but for PhysC.

655



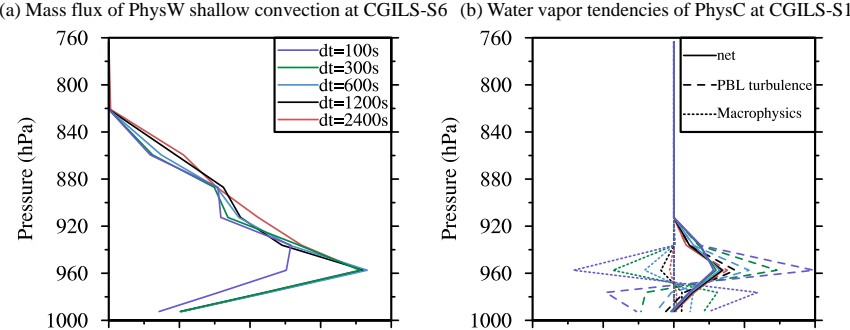

**Figure 10:** (a) Time-averaged Tiedtke-Bechtold shallow convective mass fluxes for PhysW at CGILS-S6 using each time step. (b) Time-averaged water vapor tendencies of the macrophysics (dotted) and PBL turbulence (dashed), and the net water vapor budget of PhysC (solid) for CGILS-S12 using each time step. The color indexes for each time step follow that in (a).