# Peer review of "Intercomparison of the weather and climate physics suites of a unified forecast/climate model system (GRIST-A22.7.28) based on single column modeling"

_Geoscientific Model Development, 2022_

## Author Comment (AC1)

**Response to Reviewer #2**

RC2: 'Comment on gmd-2022-283':

This paper leverages the single column modeling framework to compare two physics suites in a unified forecast/climate modeling system and performs various sensitivity experiments. I felt the paper was well written and I do recommend publication after the following issues are addressed.

**Response:** The authors thank this reviewer for many helpful suggestions.**

1. I felt that all (line) plots could benefit from adding a background grid for more ease in interpretation of results (though this is more of a personal preference and leave it up to the authors). **Response:** Done. All the line plots are added with tick marks and labels at X-Axis top and Y-Axis

right. The legends for the line plots are also modified to help clearly showing the results.

2. It is my viewpoint that SCMs are very useful (even vital) tools for GCMs. However, including a brief discussion on their limitations would be useful for the reader.

Response: Done. We have added the limitation of SCM in the Introduction (Lines 86-89).

"The limitation of SCMs lies in the absence of (3D) physics-dynamics interaction. In the cases such as propagating rainfall episodes or middle-latitude cyclones, the SCM may be viewed only as a way to describe a constrained balance of the model physics to the prescribed large-scale condition (Zhang et al. 2016)."

3. Figure 1: Would it be possible to add profiles of the cloud ice?

**Response:** Done. The time average cloud ice mixing ratio for convection active and suppressed periods is added in Figures 3e and 3f. PhysW produces larger cloud ice content than PhysC because of the stronger vertical transport by the TB deep convection.

4. For TWP-ICE is it possible to show the profiles of Temperature and Moisture errors with respect to observations?

**Response:** Done. We add a figure (Figure 2) to show the temperature and water vapor errors for TWP-ICE. The modeled temperature and moisture are not nudged towards the observation during integration. Cool and dry biases increase after day 6 when the large-scale forcing weakens. In the convection suppressed period, PhysW shows slightly smaller negative errors of temperature as compared with PhysC.

5. Some figures use Pressure for the y-coordinate while some use kilometers/meters. Please use the same coordinate for all plots so that they can be directly compared to another.

**Response:** The y-coordinate (pressure or height) of a plot depends on the reference data (IOP observation or LES). The pressure coordinate is used for the deep convection case (TWP-ICE) while height coordinate is used to evaluate low clouds in the DYCOMS and CGILS cases.

6. Regarding the DYCOMS case, it was not clear to me at all if the authors used the research flight 1 (RF01) or RF02. On page 6 it is stated that DYCOMS focuses on "nonprecipitating marine stratocumulus" which suggests RF01, while on page 9 it is stated that DYCOMS is "... with embedded pockets of drizzling open cellular convection" which suggests RF02. Please explicitly state in the document which flight segment was used and the appropriate reference.

**Response:** We used the research flight 1 (RF01) of the DYCOMS case (Stevens et al. 2005). The sentence on page 9 is modified as: "The DYCOMS-RF01 is a test case with steady nocturnal non-precipitating stratocumulus-topped mixed layer."

7. Figure 3: Please add the LES mean and spread for variables where available from the LES intercomparison study. Whether it was for RF01 or RF02 this data is publicly available and would add a nice reference point.

**Response:** Done. We add the LES data for reference of the DYCOMS-RF01 modeling (which is available at http://gcss-dime.giss.nasa.gov). The LES mean and spread for cloud liquid water mixing ratio are added to Figure 5. In addition, the LES data shows that the fraction columns with cloud present is ~0.92, representing a reference of cloud fraction.

8. Regarding the DYCOMS results... 30 layer vertical resolution is quite coarse for this regime. I feel like the paper would be strengthened by adding a vertical resolution sensitivity for the DYCOMS case. Often times parameterizations are tuned to achieve optimal results for stratocumulus to compensate for these very coarse vertical resolutions; which often breaks down when the vertical resolution is increased to something more appropriate for this regime. This would be a nice way to exploit any potential sensitivities to vertical resolution for each physics package and the SCM is the ideal vehicle to do this.

**Response:** Done. A vertical resolution sensitivity experiment is conducted for the DYCOMS case by increasing vertical resolution from 30 full model layers to 60 layers (section 5.3 on page 14). The increased levels halve the distance between the default model levels.

Figure 15 (in the revised manuscript) compares the temperature and cloud properties for the runs using 60 layers (referred to as "60levs") and 30 layers (referred to as "30levs"). The cloud liquid water content for PhysW decreases by ~50% as the vertical resolution increases. This is accompanied by a lifted inversion and cloud base. The water vapor budget illustrates that the shallow convection for PhysW strengthens with the increasing resolution and transports water vapor to higher levels (Figure 16). It implies that the collaborative effect of shallow convection and PBL turbulence in PhysW to produce stratocumulus clouds is sensitive to vertical resolution. In contrast to PhysW, PhysC has nearly identical cloud properties in the 60levs and 30levs runs, showing a mild sensitivity to vertical resolution.

9. Figure 6, I feel like panels e) and h) should be made into their own figure. There are shared and conflicting color schemes with the other plots in this panel that makes it very confusing (and easily misleading) to interpret.

**Response:** Done. Panels e) and h) are plotted as Figure 9.**

10. Page 11, lines 314-315. The authors state "the nocu runs of PhysC and PhysW produce highly consistent precipitation evolution...". This is really hard to tell from this plot to my eye. I think it would be more illuminating to show the evolution of the difference between the two nocu runs and the two base runs.

**Response:** Done. The figure is modified to show absolute differences of precipitation evolution between the two "nocu" runs and the two base runs (Figure 8a). The absolute difference of precipitation between the "nocu" runs is smaller than that of the base runs in the convection active period (the first 6 days). The difference in the water vapor budgets between the PhysC and PhysW "nocu" runs is also smaller (comparing Figure 4 and Figure 9).

During the convection suppressed period with weakened large-scale forcing, the difference between the "nocu" runs is larger than that of the base runs from day 7 to day 10. This is mainly attributed to the peak time difference of weak rainfall events.

11. Figure7, is there an observational source available to plot here?

**Response:** Done. The observed cloud fraction is added in this figure (Figure 10 in the revised manuscript).

12. In the conclusions section it would be nice if the authors could expand on how this work would more broadly benefit modeling centers.

**Response:** Done. We have now provided some concluding remarks in the last two paragraphs of this paper.

---

## Author Comment (AC3)

**Response to Reviewer #1**

RC1: 'Comment on gmd-2022-283':

General comments

In this manuscript, two physics suites separately coupled in a unified weather-climate model are described. Three field cases are used to understand the difference in model behaviors between the two physics suites within single column model configuration. The authors provide evidence for contribution of convective parameterization scheme to the major discrepancy in the simulated precipitation and clouds. Model sensitivity to time step is attributable to the interaction between microphysics and other processes. Although the discrepancies between the simulations are clearly illustrated, the underlying reasons need further investigation as suggested in the specific comments. Meanwhile, the roles for the two physics suites in unified weather and climate modeling need to be clarified to provide reference for other modeling centers. I would recommend it for publication in GMD after minor revisions in terms of the specific issues below.

**Response:** The authors thank this reviewer for many helpful suggestions.

Specific comments

L40 and L46: Please briefly describe the distinct formulation of unified weather-climate modeling (e.g., GRIST) that is different from other weather and climate models.

**Response:** The development of GRIST boosts the creation of a new model architecture that facilitates unified weather and climate modeling. In practice, because global weather and climate modeling differ significantly in terms of their spatial and temporal scales, the unification is realized by maximizing the possibility of constructing weather and climate models using a single model framework and dynamical core. GRIST further pursues to maximize the possibility of using a unified model formulation with minimum application-specific changes for weather-to-climate forecast applications.

L133: The statement "… a clear difference…, that is, dynamics and all the microphysical processes are more closely coupled together" could be more specific. A schematic flowchart of the computational procedure, if possible, could be beneficial to illustrating the difference in coupling strategies of dynamics and physics between PhysC and PhysW.

**Response:** Done. We add a figure (Figure 1) to show the different coupling strategies and process orders for the two physics suites.

The physical processes of PhysC are sequentially coupled with an order from the wet (deep convection, shallow convection, stratiform cloud condensation, and cloud microphysics) to dry (radiation transfer, surface flux, and PBL turbulence) processes. In contrast, PhysW adopts a coupling order from fast to slow processes. Cloud microphysics is computed first, and it is not specifically tied to those physical assumptions related to large-scale stratiform-like clouds. PBL turbulence, cumulus convection, and radiation transfer are called after the atmospheric state updated by the microphysics. Deep and shallow convection share the same parameterization, but they do not co-occur within one time step.

L172: The mentioned citation for DYCOMS experiment (Table 1) is missing in the reference list. In addition, please check the long name of the experiment in Table 1. And the location (lat, lon) should be precise with direction units.

**Response:** We have modified Table 1 and complemented the reference.

L221: It is found that the ratio of convective precipitation in PhysC is quite smaller during day 2-4 than that during day 0-2 at the peak value time of each event (Figure 1a). What background

environment or treatment in model physics contributes to the difference? Does the difference has influence on the mean state of cloud profile during the convection active period (Figure 1c, e)?

**Response:** The closure of the parameterized convection for PhysC is based on the convective available potential energy (CAPE). That is, an equilibrium is assumed between the convection (reducing CAPE) and the grid-scale environment (generating CAPE). Due to the large consumption of CAPE at the first two days, the convective precipitation decreases afterwards.

Figure R1 shows the time-averaged cloud fraction for days 0-2 and days 2-4, respectively. The pattern of cloud profiles modeled by PhysC overall resembles the observation. PhysC produces ~0.3 more middle and high clouds than that observed at day 2-4.

[Figure]

**Figure R1:** Time-averaged cloud fraction of days 0-2 and days 2-4 for PhysC and IOP observation.

L272: DYCOMS shows consistent marine stratocumulus cloud amount generated by different interactions of the physical processes in PhysW and PhysC. In CGILS, however, difference in stratocumulus cloud fraction at S11 between PhysW and PhysC is apparent. Is this difference primarily attributed to the shallow convection or other processes such as PBL turbulence?

**Response:** We add a sensitivity experiment for CGILS to examine the role of PBL turbulence on stratocumulus (Supporting Information). The UW wet turbulence scheme is coupled to the PhysW suite and replaces the YSU scheme. Figure S1 compares the time-averaged cloud fraction and cloud liquid water mixing ratio simulated by PhysW (UW) and PhysW (YSU), respectively. Compared with PhysW (YSU), PhysW (UW) increases stratocumulus at CGILS-S11 that more resembles the observation. The UW wet turbulence scheme increases the moisture transport and thus reduces the ventilation of shallow convection (Figure S2). This leads to condensation instead of evaporation occurring in the microphysics. Comparing Figure 7b and Figure S2, shallow convection in PhysW (UW) is still more active than that in PhysC. It confirms that the collaborative effect of shallow convection and PBL turbulence is the key difference of PhysW to PhysC for stratiform cloud process. Changes in the PBL turbulence scheme can notably impact this collaborative effect, leading to improved stratocumulus for PhysW.

L274: Please clarify the "lower levels" with specific pressure layers.

**Response:** The "lower levels" means a height of 600-1000 m. The specific depth is added in the sentence.

L323: Compared to the convection active period, the increment of middle and low clouds (500-900 hPa) without convection parameterization seems larger in PhysW during the convection suppressed

period. Why does the increment become larger when the convection is suppressed rather than active?

**Response:** The large-scale forcing and precipitation are both weaker in the convection suppressed period than that in the convection active period. Without the vertical transport of convection, moisture tends to accumulate and easily reach saturation. Thus, condensate increases in middle and low levels (below 500 hPa), leading to notable increment of clouds. This increment in clouds is also seen in the "nocu" runs of PhysW for the CGILS case. Their physical mechanisms are consistent.

L417: What roles do the two physics suites play in the current unified weather and climate modeling? The authors may consider linking the main conclusion to future implication of this study for other modeling centers.

**Response:** PhysW is more optimal for kilometer-scale simulations due to its physical and computational performance. PhysC is, in principle, more optimal for decades-to-centuries scale long-term climate simulations. The two physics suites can do some common work. For instance, PhysW can produce a realistic model climate with close-to-zero top-of-atmosphere radiation budget and well captures the global and regional precipitation patterns. But its net cloud radiative forcing needs further improvement. Looking to the future, it is our hope that a unified model physics suite with minimum application-specific changes can be used (and behaves properly) for both weather and climate modeling. We have provided some concluding remarks in the last two paragraphs of this paper.

Technical corrections

L90: Section 5 explores

**Response:** Modified.

L142: while using

**Response:** Modified.

L156: subscript phys

**Response:** Modified.

L183: Please check the vector symbol in the equation (4).

**Response:** It is modified as $\vec{V} \cdot \nabla q$.

Please make sure the reference list includes all the citations in the manuscript.

**Response:** Thanks for your suggestion. the reference list has been complemented.